# The Role of Oxidative Stress and Inflammatory Parameters in Heart Failure

**DOI:** 10.3390/medicina60050760

**Published:** 2024-05-02

**Authors:** Karolina Wróbel-Nowicka, Celina Wojciechowska, Wojciech Jacheć, Marzena Zalewska, Ewa Romuk

**Affiliations:** 1Medical Laboratory of Teresa Fryda, Katowice, Laboratory Branch in Specialist Hospital in Zabrze, 10, M.C-Skłodowska St., 41-800 Zabrze, Poland; karolina.wrobel@diag.pl; 22nd Department of Cardiology, Faculty of Medical Sciences in Zabrze, Medical University of Silesia, 10, M.C-Skłodowska St., 41-800 Zabrze, Poland; cwojciechowska@sum.edu.pl (C.W.); wjachec@sum.edu.pl (W.J.); 3Department of Basic Medical Sciences, Faculty of Public Health in Bytom, Medical University of Silesia, Piekarska St., 41-902 Bytom, Poland; mzalewska@sum.edu.pl; 4Department of Biochemistry, Faculty of Medical Sciences in Zabrze, Medical University of Silesia, 19, Jordan St., 41-808 Zabrze, Poland

**Keywords:** heart failure, oxidative stress, inflammation, biomarkers

## Abstract

Heart failure (HF) remains a major medical and social problem. The NT-pro-brain natriuretic peptide (NT-proBNP) and its active form, brain-type natriuretic peptide (BNP), in a simple blood test are the gold-standard biomarkers for HF diagnosis. However, even good biomarkers such as natriuretic peptides fail to predict all the risks associated with HF due to the diversity of the mechanisms involved. The pathophysiology of HF is determined by numerous factors, including oxidative stress, inflammation, neuroendocrine activation, pathological angiogenesis, changes in apoptotic pathways, fibrosis and vascular remodeling. High readmission and mortality rates prompt a search for new markers for the diagnosis, prognosis and treatment of HF. Oxidative-stress-mediated inflammation plays a crucial role in the development of subsequent changes in the failing heart and provides a new insight into this complex mechanism. Oxidative stress and inflammatory biomarkers appear to be a promising diagnostic and prognostic tool in patients with HF. This systematic review provides an overview of the current knowledge about oxidative stress and inflammation parameters as markers of HF.

## 1. Introduction

HF remains one of the biggest health issues facing people around the world. The increasing numbers of new HF cases and high readmission and mortality rates have created a major medical and social problem. The increased prevalence of HF is due to the aging of the population, improved survival after myocardial infarction and increasing prevalence of cardiovascular risk factors [1]. HF is considered a result and the final stage of chronic diseases related to the circulatory system. Three-quarters of HF patients are diagnosed with hypertension, and many of them have ischemic heart disease, heart attacks, arrhythmia or heart defects. All these situations lead to damage to the heart muscle and to the insufficient pumping of blood to supply the tissues [2,3,4].

The gold standard for detecting HF is a simple blood test for NT-pro-brain natriuretic peptide (NT-pro BNP) and its active form, brain-type natriuretic peptide (BNP). These markers were discovered in 1988 and are now recommended in the guidelines of the European Society of Cardiology (ESC) Clinical Practice Guidelines and the American (AHA/ACC/HFSA) Guidelines for the assessment of HF [5,6].

In clinical practice, NT-pro BNP is a better marker than BNP because of the lower biological diversity and longer plasma half-life [5]. The usefulness of NT-pro BNP for risk stratification varies depending on the stage of HF, time of assessment and duration of follow-up. However, there is no conclusive evidence that the plasma NT-pro BNP concentration is a guide for more effective therapy [7].

Even good biomarkers such as natriuretic peptides fail to predict all the risks associated with HF. This is due to the enormous diversity of mechanisms involved in the development of HF. A large number of clinical and experimental studies prove that oxidative stress and the activation of inflammation plays a crucial role in the pathophysiology of HF. Increased oxidative stress and inflammation can be classified as new risk factors for cardiovascular diseases. There is a need for a new tool to improve the effectiveness of diagnosis, prognosis and treatment of patients with no HF symptoms.

Oxidative stress and inflammation contribute to HF’s development and progression. These mechanisms are responsible for cardiac remodeling in HF. Changes affect intracellular signaling pathways, the redox state of the cell, proliferation of fibroblasts and activation of intracellular matrix metalloproteinases, leading to changes in the structure of the extracellular matrix and to hypoxia [8,9].

## 2. Hypoxia and Heart Metabolism

In adult hearts, chronic exposure to hypoxia enhances the adaptive capacity of the heart, particularly its glucose metabolism, offering protection against acute hypoxia. In heart failure, reduced oxygen use leads to metabolic remodeling, referred to as a “fuel-depleted engine,” contributing to the complex changes observed in affected patients [10]. Hypoxia, a condition characterized by a lack of oxygen, plays a central role in the development and metabolic changes seen in HF [10]. During cardiac development, the heart initially operates in a hypoxic environment and undergoes metabolic shifts as it transitions from a hypoxic to a normoxic state. However, HF leads to pathological metabolic remodeling, involving changes in substrate utilization, increased oxidative stress and even the re-expression of fetal-associated genes. During acute stress, the heart shifts from a “glucose-fatty acid cycle” to prioritize carbohydrate catabolism to meet adenosine-5-triphosphoric acid (ATP) demands. However, persistent stress, such as chronic hemodynamic overload, triggers a shift from an adult to fetal metabolic phenotype, resulting in extensive metabolic remodeling. This adaptive switch, initially beneficial, ultimately leads to insulin insensitivity and a loss of metabolic flexibility. Some studies addressing the timing of cardiac energetic impairment in HF suggest that it occurs only in advanced stages. However, pioneering studies using 31P NMRs (Phosphorus-31 nuclear magnetic resonance) identified a reduction in myocardial energy reserves as a potent prognostic indicator in HF, even when ATP levels seem to be initially maintained. Mitochondrial dysfunction, including reduced density and disrupted structure, intensifies oxidative damage, leading to cardiomyocyte loss. Interestingly, the relationship between the sodium–potassium pump function and metabolism is bidirectional. While declining ATP levels might affect ion transport, evidence suggests that NKA inhibition and elevated Na may influence the mitochondrial metabolism [11,12]. This process is guided by hypoxia-inducible factors (HIFs), with HIF-1α being vital for embryonic survival and metabolic adaptations during development [10,12]. HIF-1α is a master regulator of oxygen homeostasis, responding to varying oxygen levels indirectly through enzymes such as prolyl hydroxylases and asparagine hydroxylase. In hypoxia, these enzymes are inhibited, allowing HIF-1α to move to the nucleus, where it pairs with other molecules to regulate the hypoxia response genes [11]. HIF-1α, responding to hypoxia, plays a crucial role in modulating inflammatory processes, particularly in monocyte differentiation to macrophages and subsequent cytokine production. Macrophages, in turn, release various cytokines, fostering a positive feedback loop between tumor necrosis factor α (TNF-α) and HIF-1α, bolstering monocyte differentiation. Controversies exist regarding the effects of monocyte/macrophage actions and HIF-1α in cardiac remodeling under hypoxia. While some studies suggest that HIF-1α-induced macrophage accumulation suppresses heart fibrosis through cytokine OSM (Oncostatin M), which might have beneficial effects, other research indicates that OSM produced by macrophages contributes to inflammatory processes and cardiomyocyte remodeling. Chronic and intermittent hypoxia upregulates several HIF-1α target genes, leading to the increased expression of molecules, such as inducible nitric oxide synthase (iNOS), endothelin-1 (ET-1) and angiotensin-converting enzyme (ACE). While iNOS has a protective role against oxidative stress and apoptosis in the heart, ET-1 and ACE are associated with cardiac hypertrophy. However, the direct relationship between these molecules and hypobaric hypoxia-induced right ventricular hypertrophy (RVH) remains unclear. Nonetheless, studies in other types of hypoxia have shown that angiotensin II receptor blockers reduce RVH, suggesting a potential link [11,13]. Furthermore, chronic hypoxia significantly contributes to right heart failure (RHF) through factors such as pulmonary hypertension and cardiovascular comorbidities. This leads to RVH and metabolic changes, including the increased production of reactive oxygen species (ROS) [12,13]. These findings emphasize the importance of considering hypoxia in the cardiac metabolism and its implications in both HF and related conditions, offering potential therapeutic insights.

## 3. Hypoxia and Mitochondrion

Mitochondria, the cell’s energy producers, are also involved in hypoxia and can trigger oxidative stress. Mitochondrial DNA (mtDNA) is especially vulnerable, potentially leading to reduced cellular respiration and ATP production. Mitochondria play a pivotal role in the intricate web of oxidative stress, having a profound impact on heart health. Often referred to as the “powerhouse of the cell”, these organelles serve a dual purpose in both generating and mitigating reactive oxygen species (ROS) [14,15]. Under normal physiological conditions, mitochondria primarily produce ROS as byproducts of the electron transfer process in the mitochondrial respiratory chain, and complexes I and III are prominent contributors. Proteins, such as p66shc, monoamine oxidases (MAOs) and NADPH oxidase (NOX4), also add to the pool of mitochondrial ROS. These ROS, including superoxide anion and hydrogen peroxide, typically function as vital signaling molecules for processes such as heart development and cardiomyocyte maturation. However, when the delicate balance of ROS production is disrupted, oxidative stress ensues [14,15,16]. In the case of ischemia–reperfusion injury (IRI), the temporary lack of blood flow during myocardial ischemia triggers mitochondrial dysfunction. Subsequent reperfusion exacerbates ROS production, ultimately resulting in tissue necrosis. Mitochondrial damage, including the opening of the mitochondrial permeability transition pore (MPTP), plays a crucial role in IRI-induced myocardial injury [14,15]. In HF, chronic oxidative stress is linked to an elevation in ROS levels through proteins such as NADPH oxidase 2 (NOX2) and NOX4. These ROS contribute to cardiac hypertrophy and remodeling, ultimately compromising cardiac function. Research into the precise role of NOX4 in HF is ongoing, highlighting the intricate interplay of mitochondrial ROS in cardiac pathophysiology [14,16]. Maintaining mitochondrial quality is achieved through two primary pathways: the ubiquitin–proteasome system (UPS) and mitophagy. The UPS pathway degrades damaged mitochondrial proteins, while mitophagy selectively removes entire dysfunctional mitochondria, safeguarding overall mitochondrial health [15,16]. Recent research has uncovered additional mechanisms through which mitochondria contribute to the pathophysiology of HF. These mechanisms include metabolic flux bottlenecks, redox imbalances, protein modifications, disruption of mitochondrial calcium homeostasis and inflammation. Each of these mechanisms underscores the multifaceted role of mitochondria in cardiac health and their significance in maintaining the delicate balance between ROS signaling and oxidative stress [16,17]. In conclusion, mitochondria, with their dual role in both ROS generation and mitigation, have a profound impact on oxidative stress and its implications for cardiac health. The balance between ROS generation and scavenging within these organelles is a critical determinant of cardiac physiology and pathophysiology [14,16,17]. Oxidative stress refers to an imbalanced condition in which the production of reactive oxygen and nitrogen species (ROS/RNS) exceeds the body’s capacity to detoxify these harmful molecules. This imbalance can result in damage to various cellular components and tissues, including the mitochondria, which serve as the energy-producing organelles within cells [9,18]. Abnormalities in ryanodine receptor (RyR2) activity are implicated in HF development. Dysfunctional RyR2 can cause diastolic Ca^2+^ leaking, reducing contractile force generation. These issues may result from direct oxidative modifications, hyperphosphorylation by enzymes sensitive to the intracellular redox state and ROS-activated Ca^2+^/calmodulin-dependent protein kinase II (CaMKII), promoting Ca^2+^ overload [9]. Redox-induced alterations in the sarcoplasmic reticulum and Ca^2+^-ATPase activity contribute to abnormal Ca^2+^ handling and contractile failure. ROS can impair diastolic function through disulfide bridge formation within specific cardiac proteins [9,18]. ROS-induced disruptions in cellular ion homeostasis may promote ventricular arrhythmias through mechanisms, such as focal triggered activity, re-entrant circuits and early afterdepolarizations (EADs). Oxidative stress can create areas of inhomogeneous excitability, leading to ventricular arrhythmias through re-entrant circuits.

## 4. Oxidative Stress and Inflammation in Cardiac Remodeling and HF

Oxidative stress and inflammation are interdependent conditions that occur simultaneously. Oxidative stress products enhance pro-inflammatory responses, whereas inflammatory cells release ROS at the site of inflammation and promote oxidative damage [10].

ROS attract circulating inflammatory cells through multiple mechanisms, such as upregulation of chemokines, activation of neutrophil integrins and expression of surface adhesion molecules by endothelial cells [9].

Oxidative stress and inflammation are responsible for the activation of signaling molecules in myocardial remodeling and failure. ROS activate a variety of hypertrophy signaling kinases and transcription factors. Among others, it activates tyrosine kinase Src, GTP binding protein Ras, protein kinase C, mitogen-activated protein kinases (MAPKs) and Jun-nuclear kinase (JNK) [19,20,21].

ROS-generating enzymes such as NOX 2 contribute to hypertrophy [9,18]. The NADPH oxidases (NOXs), abundantly present in the vascular system, serve as pivotal enzymes in generating ROS. They facilitate the conversion of oxygen into a superoxide anion by utilizing an electron from NADPH. These enzymes, consisting of catalytic molecules known as Nox, have five identified isoforms, notably Nox1, Nox2 and Nox4, which hold significance in the pathophysiology of HF, reliant on the presence of p47phox and the GTPase. Nox-4, primarily localized to mitochondria in cardiac myocytes, plays a crucial role in increasing ROS production and subsequent cardiac remodeling due to factors such as pressure overload and aging. NADPH oxidase activity amplifies significantly in response to various triggers linked to the pathophysiology of HF, including mechanical stretch, angiotensin II, ET-1 and TNF-α [18,22,23,24,25,26,27,28].

Nitric oxide synthases (NOSs), essential for nitric oxide (NO) synthesis, are involved in HF, potentially producing harmful O^−2^ instead of NO, increasing ROS generation. RNS, derived from nitric oxide and superoxide through enzymatic activity, play a role in oxidative stress and can lead to the formation of free radicals [7,22,28]. Under pathological conditions, NOS can become uncoupled, resulting in the generation of superoxide (O^2−^) as a byproduct of nitric oxide synthesis, often associated with electron leakage during mitochondrial respiration [7].

These changes activate the process of apoptosis, another important contributor to remodeling and dysfunction, induced by ROS-mediated DNA and mitochondrial damage as well as the activation of proapoptotic signaling kinases [23].

In the next step, ROS lead to the activation of the nuclear enzyme poly (ADP-ribose) polymerase-1 (PARP-1). PARP-1 regulates the expression of inflammatory mediators, which facilitate the progression of cardiac remodeling. ROS also stimulate transcription factors such as nuclear factor-_k_B and activator protein-1 to stimulate matrix metalloproteinase (MMP) expression [24].

MMPs play a pivotal role in normal tissue remodeling processes, such as cell migration, invasion, proliferation and apoptosis. The activity of MMP has been shown to be increased in failing hearts [25].

The increased activity of MMPs resulting from excessive ROS production might influence the structural properties of the myocardium by providing an abnormal extracellular environment that interacts with myocytes. Oxidative stress and inflammation also affect contractile function by modifying proteins involved in excitation–contraction coupling [26].

The right ventricle (RV) exhibits heightened vulnerability to oxidative stress in comparison to the left ventricle (LV). This increased susceptibility in the RV is partly attributed to its inability to regulate the expression of manganese superoxide dismutase, which contributes to oxidative stress in the RV [7,27,28]. Oxidative stress plays a pivotal role in pulmonary vascular remodeling, with monocytes accumulating in pulmonary arterioles, leading to ROS generation. This accumulation of ROS contributes to an elevated presence of free radicals in the context of pulmonary hypertension (PH) [7,28].

## 5. Antioxidant Defense

The endogenous defense mechanisms against reactive oxygen species (ROS) encompass a range of vital components. These include superoxide dismutase (SOD), catalase, glutathione peroxidase (GPx), nicotinamide adenine dinucleotide (NAD+) and glutathione (GSH). In HF, a notable feature is the reduction in the activity of these antioxidant enzymes, especially SOD, catalase and GPx, resulting in elevated oxidative stress [9,18,26]. In HF models, diminished levels of NAD+ and GSH have been associated with worsened outcomes, as they are pivotal in redox reactions and the detoxification of ROS. Encouragingly, supplementation with NAD+ precursors, such as nicotinamide riboside (NR), has demonstrated significant potential in enhancing cardiac function and redox equilibrium in HF [18,26].

SODs are metalloproteinases with a pivotal role in transforming superoxide anions (.O^2−^) into hydrogen peroxide (H_2_O_2_). These highly effective antioxidant enzymes detoxify superoxide anions, preventing their reaction with nitric oxide (NO) and the formation of peroxynitrite (ONOO-). SODs encompass three isoforms—cytosolic SOD1, mitochondrial SOD2 and extracellular SOD3—each requiring distinct metallic cofactors and dimerization for activity [26,27].

Catalase is an enzyme specializing in the detoxification of hydrogen peroxide, with its function dependent on the hydrogen peroxide concentration. At high concentrations, it primarily acts as a catalytic detoxifier, while at lower concentrations, it predominantly functions as a peroxidase. Catalase exists as a tetrameric heme protein [26,27].

GPxs are tetrameric selenoproteins found in the cytoplasm, nuclei and mitochondria. They are responsible for detoxifying H_2_O_2_ into water and eliminating peroxide residues from lipids. This process relies on the reducing capabilities of the glutathione/glutathione disulfide (GSH/GSSG) couple and involves the continuous reduction of GSSG to GSH, aided by glutathione reductase and NADPH generated through the pentose phosphate pathway [7,26,28]. Various non-enzymatic antioxidants also contribute to shielding cells from ROS damage. GSH is a major intracellular thiol, primarily present in its reduced form. It serves several antioxidant roles, including acting as a cofactor for Gpx, a transition metal chelator and a regenerator of vitamins C and E. Additionally, albumin is often regarded as the plasma equivalent of GSH. Vitamin C (ascorbic acid) is recognized as the most effective water-soluble antioxidant in human plasma, while vitamin E (tocopherol) helps combat lipid peroxidation due to its lipophilic nature and membrane location [26,27,28]. These antioxidant mechanisms are vital for neutralizing ROS and preserving cellular redox balance. Their proper functioning is of the utmost importance in safeguarding cells and tissues from oxidative stress and its associated damage [26,27,28].

Thiols, such as cysteine and glutathione (GSH), are of paramount significance as antioxidants, ensuring the maintenance of cellular redox balance and counteracting oxidative stress [28].

The efficacy of interventions designed to restrain ROS production, such as xanthine oxidase inhibitors (e.g., allopurinol) and NOS uncoupling inhibitors, varies across different clinical trials. Moreover, the supplementation of exogenous antioxidants, such as vitamins E, C and A, has not consistently led to improved cardiovascular outcomes in HF patients [18,26]. A. van der Pol et al. suggest that future strategies for mitigating oxidative stress in HF could concentrate on two primary approaches: augmenting endogenous antioxidant capacity through GSH and NAD+ precursor supplementation and enhancing the expression and activity of antioxidant-producing enzymes linked to the *γ*-glutamyl cycle and NAD+ production [18,26,27]. Currently, the pathophysiological concept of HF emphasizes the enormous role of oxidative parameters and inflammatory markers. It is important to isolate a panel of oxidative stress and inflammation parameters, which, on the one hand, could play a role as prognostic parameters and, on the other hand, allow for a better explanation of the presence of clinical symptoms and determine a patient’s prognosis for outcome. Most importantly, however, the panel of parameters should also affect treatment selection, which, as a consequence, would inhibit some processes that damage the heart or even improve its function.

## 6. Biomarkers in Heart Failure

The pathophysiology of HF is determined by many factors, including oxidative stress, inflammation, neuroendocrine activation, severe angiogenesis, changes in apoptotic pathways, fibrosis and vascular remodeling. The oxidative-stress-mediated inflammation seems crucial in the development of subsequent changes in the failing heart and provides a new insight into this complex pathophysiological mechanism. The simultaneous measurement of several HF biomarkers will allow us to better describe the patient’s metabolic situation and provide more personalized treatments for HF as well as estimate the future risk of cardiovascular events. Because of the unsatisfactory current therapeutic results, any attempt to find a new method for the detection and treatment of HF is worthy of attention.

The multiple-biomarker approach has been tested, but it remains unclear whether its economic burden and reproducibility are suitable for practical use [29,30,31,32]. Table 1 shows the markers we discuss in terms of their usefulness in laboratory diagnostics.

Search Strategy

A systematic search was conducted to assess the oxidative stress and inflammatory markers using the following databases: PubMed, Web of Science, MEDLINE and SCO-PUS. In the search, we were most interested in articles from the last 6 years. Thanks to this, we managed to choose the newest and most interesting markers within our interests. The literature search was performed using “oxidative stress markers and HF” (about 610 articles), “inflammatory markers and HF” (about 965 articles), “oxidative stress and inflammatory markers and HF” (about 140 articles), as well as other synonymous terms.

### 6.1. Inflammatory and Fibrosis Biomarkers

#### 6.1.1. Apelin

Apelin is a novel adipokine that acts as an endogenous ligand of the specific orphan apelin receptor (APJ). Apelin is part of the apelinergic system, which includes apelin, ELABELA (ELA) and the APJ receptor. Apelin plays a crucial role in the cardiovascular system. Under normal conditions, it enhances cardiac contractility, increases stroke volume, causes vasodilation, promotes diuresis and lowers blood pressure [33]. This system is expressed in various human tissues and organs, including the heart [33]. Apelin exists in various isoforms, with apelin-36 being a prominent one. An increase in the plasma level of apelin-36 has been associated with oxidative stress [34,35]. Apelin-36 has positive inotropic effects, i.e., enhancing the force of heart muscle contraction. It widens blood vessels too, reducing resistance and making it easier for the heart to pump blood and helps protect against the development of atherosclerosis [34,35,36,37]. Apelin-36 may be used as a biomarker of HF. In cases of HF, the plasma levels of apelin-36 increase along with signs of oxidative stress [34,36]. Ferdinal et al. proved that chronic systemic hypoxia can lead to ventricular hypertrophy, myocardial structural damage and ventricular remodeling. Apelin-36 levels were found to be elevated in this context, indicating its role in the development of HF [34]. Oxidative stress, marked by an increase in reactive oxygen species (ROS), has been linked to increased apelin levels. Apelin appears to be associated with regulating the oxidative stress response in the heart [34]. The progression of HF involves multiple pathogenic processes, including the renin–angiotensin–aldosterone system (RAAS), the sympathetic nervous system, inflammation and oxidative stress [36,37]. A study by Goidescu et al. demonstrated that the serum angiotensin-converting enzyme 2 (ACE2) and apelin-13 activity are correlated with an unfavorable clinical evolution in patients with reduced ejection fraction HF [36]. These biomarkers can be useful for evaluating the prognosis of patients and may play a role in understanding the molecular pathways involved in HF progression [35,36].

#### 6.1.2. FGF23

Fibroblast growth factor 23 (FGF-23) is a hormone primarily originating from bones. It plays a significant role in regulating the renal phosphate balance and vitamin D metabolism. In the context of HF, particularly in heart failure with preserved ejection fraction (HFpEF), FGF-23 has drawn considerable attention due to its potential influence on cardiovascular health [38]. Some studies have shown that elevated levels of FGF-23 are associated with left ventricular hypertrophy (LVH) and a poorer prognosis. This association has been observed in various heart conditions, including chronic kidney disease (CKD), ischemic cardiomyopathy and heart failure with reduced ejection fraction (HFrEF) [39,40]. However, the role of FGF-23 in myocardial abnormalities commonly seen in HFpEF, such as left ventricular diastolic dysfunction, LVH, interstitial fibrosis and right ventricular dysfunction, is not well understood [38,39,40]. In a study by Benes et al., FGF-23 emerged as one of the most significantly different proteins in the blood plasma of HFrEF patients with severe RV dysfunction. FGF-23 showed potential as a biomarker for RV dysfunction in HFrEF patients, as it exhibited a strong correlation with RV dysfunction, regardless of other factors, such as congestion. In contrast to BNP, an established HF biomarker associated with both RV and LV dysfunction, FGF-23 specifically reflected RV dysfunction. The study suggests that FGF-23 may serve as an innovative biomarker for identifying RV dysfunction in patients with HFrEF. This is particularly important because there is currently no specific biomarker for RV dysfunction, and FGF-23 could aid in the diagnosis and monitoring of this condition in HFrEF patients [40,41].

In a study by Damien Gruson et al., the FGF-23 levels were found to be significantly higher in HFpEF patients compared to controls. FGF-23 was associated with multiple risk factors, signs of cardiac dysfunction and markers of renal impairment. Notably, the FGF-23 levels were correlated with myocardial fibrosis [38,41].

#### 6.1.3. Galectin-3

Galectin-3 (Gal-3) is a protein belonging to the 35 kDa lectin family, produced by macrophages during phagocytosis. Human Gal-3 is encoded by the LGALS3 gene, located on chromosome 14, and has a chimeric structure with a single carbohydrate recognition domain (CRD) connected to an N-terminal domain and the ability to form pentamers [42,43]. Gal-3 is involved in many biological processes, including inflammation, cell growth and apoptosis. Through the differentiation of pattern cells, it participates in angiogenesis [44]. By interacting with survival-related proteins, it induces endothelial cell migration and proliferation. Gal-3 found in the cell nucleus regulates gene transcription and pre-mRNA intron excision. Gal-3, present on the cell surface, regulates the diffusion, compartmentalization and endocytosis of cell membrane glycoproteins, and it also activates and retains T lymphocytes. Gal-3 in the extracellular environment induces cell adhesion and has pro-apoptotic properties [42,43,44].

Gal-3 expression increases the inflammation and cell damage caused by stretching or squeezing cardiac muscle fibers. As a marker of fibrosis, Gal-3 also has cardioprotective effects. Two large, community-based cohort studies, “Rancho Bernardo Study”, and a report by De Boer et al. show Gal-3 as a biomarker of prognostic significance, being an important predictor of mortality and readmission in chronic heart diseases [43,44]. Adding Gal-3 to a panel of BNP and hsTroponin-I biomarkers assessed in patients with heart failure allows for a more precise determination of the possibility of adverse events, such as excessive death rate. To date, Gal-3 measurement is recommended by the 2017 Guidelines of the American Heart Association for risk stratification and prognostic evaluation in patients with HF [45].

The GAL-3 in Acute HF (GALA) study revealed that levels of Gal-3 measured on admission were predictive of mortality within 30 days and short-term outcomes with a good degree of accuracy. Nevertheless, as regards longer-term consequences, specifically 1-year mortality rates, Gal-3 exhibited a somewhat decreased predictive power in this particular context [45,46]. Recent cardiovascular health studies, including the Relaxin in Acute Heart Failure (RELAX-AHF) and the ProBNP Outpatient Tailored Chronic Heart Failure (PROTECT) trials, did not unearth any discernible predictive utility associated with Gal-3 for the prospect of a patient’s demise over the course of 6 months. The conclusions from these meticulous investigations bring into question the broader applicability of Gal-3 as a prognostic marker for intermediate-term mortality and underline the complex nature of HF prognosis [44,45,46].

#### 6.1.4. GDF 15

Differentiation factor-15 (GDF-15) is a cytokine associated with inflammation and tissue regulation, primarily found in most tissues and plasma. It appears to have a protective role in the heart based on animal models and is linked to reduced growth hormone activation in rat cardiomyocytes. GDF-15 is considered a potential biomarker for cardiovascular risk [47,48]. A meta-analysis by Jin-Wen Luo et al. suggests that elevated GDF-15 levels are associated with an increased risk of all-cause mortality in congestive heart failure (CHF) patients, especially those with ischemic causes. The predictive power of GDF-15 is more significant in CHF patients with ischemic causes, and it may be a useful predictor of adverse outcomes in stable CHF patients [48,49]. GDF-15 plays a role in oxygen-sensitive gene regulation through HIF and is involved in angiogenesis and fetal heart development in a hypoxic environment. After birth, higher oxygen levels limit HIF-1α activity, and changes in mitochondrial dynamics play a role in cardiac function. GDF-15 is produced in response to mitochondrial stress and is significant in various tissues [49,50]. GDF-15 is a versatile protein, with roles in inflammation, heart protection and mitochondrial function. It is emerging as a biomarker for assessing various cardiovascular and health conditions, including heart failure and COVID-19 [50,51].

#### 6.1.5. Klotho

Klotho is a pleiotropic protein that, in humans, is encoded by the KL gene (known as α-klotho or klotho), located on chromosome 13q12. The three klotho subfamilies are α-klotho, β-klotho and γ-klotho, of which α-klotho is most highly expressed in many organs and tissues. Klotho may exist in a membrane-bound form or in a soluble circulating form that is predominant in humans. Klotho is associated with various physiological processes in the body, including its potential role in heart diseases. Klotho protein is emerging as a key player that can influence and modulate various aspects of heart function and disease [51,52]. The klotho protein has been linked to the regulation of various cellular processes, including endothelial function, oxidative stress, inflammation and fibrosis, all of which are critical factors in the development and progression of heart diseases [51,52,53].

Klotho is found in the cells of the sinoatrial node, where its expression is probably crucial for the functioning of some ion channels that regulate the excitability of heart cells. α-Klotho modulates ion channels in the cell membrane, including Na^+^/phosphate cotransporters, Na^+^/K^+^-ATPases, Ca^2+^ channels and heart-specific potassium channels. These channels determine the electrical stimulation of the heart and repolarization and participate in the abnormal stimulation of the myocardium after hypertrophy [51,52]. The endothelium lines blood vessels from the inside, and its malfunction is a characteristic feature of many cardiovascular diseases. Klotho has been shown to have a protective effect on endothelial cells by regulating the availability and increased secretion of NO, which helps dilate vessels and regulate blood flow, thus significantly reducing the risk of atherosclerosis [51,52,53]. The klotho protein exhibits anti-inflammatory properties by inhibiting the activation of TNF-α nuclear factor kappa-light-chain-enhancer of activated B cells (NF-κB) in endothelial cells, which allows for the modulation of inflammatory cytokines that may help alleviate the inflammatory processes associated with atherosclerosis and other cardiovascular diseases [53].

Klotho counteracts oxidative stress. s-Klotho inhibits insulin and insulin-like growth factor 1 (IGF-1) signaling pathway, which increases resistance to oxidative stress, thus demonstrating anti-apoptotic and anti-aging effects [52,54]. α-Klotho inhibits angiotensin II-induced cardiomyocyte proliferation and fibrosis of cardiac connective tissue cells. Moreover, klotho inhibits the expression of FGF23 and microRNA-132 in the heart, which also limits the development of cardiac fibrosis and hypertrophy. Decreased levels of klotho potentially contribute to the development of fibrotic heart disease [51,52,54]. The klotho protein, originally associated with aging and longevity, is emerging as a potential biomarker for heart disease due to its involvement in critical cellular processes of the cardiovascular system. Monitoring klotho levels may aid in the early detection, risk assessment and treatment of heart disease; however, further research and clinical trials are needed to determine its usefulness and diagnostic accuracy.

#### 6.1.6. Collagen

Myocardial fibrosis, a key player in the structural transformation of the heart during HF, has come under intense scrutiny as researchers explore molecules that could serve as biomarkers for myocardial fibrosis. Myocardial fibrosis occurs when the production of collagen, particularly types I and II, outpaces its degradation. This imbalance gives rise to two distinct forms of fibrosis: macroscopic replacement fibrosis, often seen in post-ischemic scars, and microscopic reactive fibrosis, affecting the interstitial and perivascular spaces. The latter is frequently associated with chronic conditions, such as hypertension, aortic stenosis, cardiomyopathies and other chronic causes of HF, where collagen type I fibers outnumber type III [55,56,57]. Traditionally, the gold standard for evaluating myocardial fibrosis, both qualitatively and quantitatively, has been myocardial biopsy. This invasive procedure has been used to correlate candidate biomarkers with histological evidence of fibrosis. Interstitial fibrosis, characterized by the abnormal accumulation of collagen, has been linked to LV systolic and diastolic dysfunction, arrhythmias and an increased risk of sudden cardiac death, making it a potential predictor of therapy response in HF [56,57]. Among the array of circulating biomarkers, three have stood out for their correlations with histological myocardial fibrosis: serum carboxy-terminal propeptide of procollagen type I (PICP), serum amino-terminal propeptide of procollagen type III (PIIINP) and serum collagen type I telopeptide (CITP) [57]. PICP, originating during the extracellular conversion of type I procollagen into type I collagen, has been found elevated in HF patients. It is in accord with the degree of dysfunction in patients with a reduced ejection fraction and is linked to mortality in both HF with preserved ejection fraction and HF with reduced ejection fraction. Moreover, PICP has been associated with the occurrence of ventricular arrhythmias in advanced HF and demonstrates responsiveness to treatments involving drugs such as loop diuretics and mineralocorticoid receptor antagonists [56,57]. PIIINP, arising from the extracellular conversion of type III procollagen to type III collagen, correlates with the volume of myocardial tissue replaced by collagen III fibers, known as the myocardial collagen III volume fraction (CIIIVF), measured using specific imaging techniques. Notably, treatment for HF with spironolactone has been linked to a concomitant reduction in the extension of CIIIVF and the concentration of PIIINP. Recent findings also underline the correlation between serum PIIINP levels and the outcomes and severity of HF [55,56,57]. Recent research has shed light on the potential significance of a biomarker called procollagen type III N-terminal peptide (PIIINP) in understanding and managing HF, particularly in African Americans. Mansour et al. provide unique insights into the role of PIIINP and its implications for personalized treatment strategies in this specific population [56,57]. PIIINP is a biomarker closely associated with cardiac fibrosis, a key factor in the progression of HF. The deposition of collagen in the myocardium, a process known as cardiac fibrosis, can lead to structural damage and impaired heart function. Understanding the role of PIIINP in cardiac fibrosis is critical in devising effective strategies for diagnosing and treating HF [55,56,57]. One of the key findings of the study was the correlation between PIIINP levels and genetic ancestry. PIIINP levels were inversely correlated with West African ancestry, suggesting that genetic influences play a role in the development of cardiac fibrosis in African Americans [56]. The discovery that PIIINP levels are linked to mortality risk suggests that PIIINP could be a valuable biomarker for identifying African American heart failure patients who may benefit from additional therapies aimed at combating fibrosis. This could potentially improve patient outcomes and enhance personalized treatment strategies [55,56].

#### 6.1.7. sST2

Soluble Suppression of Tumorigenicity 2 (sST2) is a member of the interleukin-1 (IL-1) receptor family, with two primary forms: cellular (ST2L) and soluble (sST2). ST2L acts as a receptor for the cytokine interleukin-33 (IL-33), providing cardioprotection by regulating the heart’s response to stress and injury. Conversely, sST2 acts as a decoy receptor, undermining this protection, and elevated sST2 levels are associated with increased mortality and risk in HF patients. Monitoring sST2 can be instrumental in patient management and evaluating HF treatments [46,58,59]. The ST2-R2 score, which includes sST2, can predict LV reverse remodeling and all-cause mortality. sST2 levels are also independently associated with atrial fibrillation in HF patients, suggesting its usefulness as a marker for recurrent atrial fibrillation [58,59]. Despite robust evidence of its prognostic value, the 2022 HF management guidelines, in contrast to the 2013 guidelines, do not recommend sST2 as a biomarker. The European Society of Cardiology guidelines take a similar stance [46,58]. In acute HF, sST2 plays a pivotal role in prognosis, especially in assessing the risk of fatal outcomes. It serves as a valuable predictor of hospital readmission due to HF when measured at discharge [46,55]. Studies, such as ASCEND-HF, MADIT-CRT and Val-HeFT, have demonstrated that serial measurements of sST2 provide valuable insights into prognosis. Elevated sST2 levels, both at admission and persisting throughout hospitalization, are associated with a higher long-term mortality risk [46,55]. A study by Pascual-Figal et al. established that in chronic heart failure, sST2 has independent prognostic value. Elevated levels are linked to the discontinuation of HF therapy, sudden cardiac death and adverse outcomes in patients with left ventricular systolic dysfunction. However, further research is needed to fully understand its advantages compared to other cardiac-specific markers [46,55,58].

### 6.2. Oxidative Stress Biomarkers

#### 6.2.1. Malondialdehyde

Malondialdehyde (MDA) is a marker of lipid peroxidation secondary to increased oxidative stress. In the context of HF and its assessment, the MDA concentration is used to evaluate the extent of lipid peroxidation, which is a process where cell membrane lipids are damaged by oxidative stress, leading to the formation of MDA as a byproduct [60,61,62]. MDA is described as a low-molecular-weight aldehyde that is formed by free radical attacks on polyunsaturated fatty acids. It is used as a biomarker of oxidative damage [61,62]. In a study by Wojciechowska et al., MDA was found to be an independent risk factor for death or heart transplantation in patients with ischemic heart failure (ICM). In nonischemic heart failure (nICM), the MDA/PSH (sulfhydryl groups) ratio has been identified as a risk factor for death or heart transplantation. Higher levels of MDA were associated with adverse outcomes [61,62,63]. Romuk et al. evaluated the importance of oxidative stress biomarkers in predicting prognosis, including death and a combined endpoint of heart transplantation or death, in HF patients. Out of a number oxidative stress markers under study, MDA was found to be an independent predictor of both death and the combined endpoint (heart transplantation or death). Higher serum MDA levels were associated with a 103% increased risk of death and a 100% increased risk of the combined endpoint per μmol/L. The results suggest that an oxidative stress marker such as MDA may play a crucial role in the pathophysiology of HF that is independently related to poor prognosis [46,61,62,63].

#### 6.2.2. Sulfhydryl Groups

SHs are produced in the body from endogenous sources, such as cysteine, and are synthesized by enzymes such as cystathionine γ-lyase (CSE) and 3-mercaptopyruvate sulfurtransferase (3-MST).

The heart is considered a significant source of SH, and changes in its metabolism can have implications for heart function. SH interacts with NO and has a strong bond with it. SH has cardioprotective effects and contributes to vasodilation, reductions in blood pressure and a decrease in heart tissue injury. It plays a role in myocardial fibrosis and inhibits the transforming growth factor-β1/drosophila mothers against the decapentaplegic protein (TGF-β1/Smad3) signaling pathway, reducing collagen deposition in the heart, which can lead to HF. SH is involved in reducing oxidative stress and inflammation, both of which are detrimental to heart tissue. SH can also protect mitochondrial function, reduce apoptosis in heart muscle cells and have antioxidant effects, which are critical for preventing heart damage during ischemia/reperfusion injury. Bajic et al. suggest that a high homocysteine concentration (HHcy) can impair the synthesis of SH, potentially contributing to cardiovascular problems [61,64,65].

SH, represented as protein sulfhydryl groups (PSH), are integral components of proteins, contributing to the stability and proper function of various biomolecules in the body. PSH can be easily oxidized by ROS, and, once oxidized, they are less effectively reduced in the bloodstream when compared to their intracellular counterparts. Consequently, the depletion of free thiols in the bloodstream serves as an indicator of a relatively stable systemic redox balance. Wojciechowska et al. investigated the prognostic value of PSH in HF patients, comparing those with ICM and nICM. The study revealed that PSH did play a prognostic role, although there were differences between the two patient groups. Specifically, in patients with ICM, PSH did not independently predict the study endpoint (comprising all-cause mortality or urgent heart transplantation). Conversely, in patients with nICM, the MDA/PSH ratio emerged as an independent risk factor for the study endpoint. These findings highlight the importance of considering PSH in a broader spectrum of redox biomarkers when assessing their role in predicting outcomes in HF patients. Notably, their prognostic relevance differs significantly between patients with ischemic and nonischemic cardiomyopathy [61,64,65,66,67].

#### 6.2.3. Uric Acid

Uric acid is the end product of purine metabolism and has both protective and detrimental effects. Xanthine oxidoreductase (XO) is an enzyme involved in purine metabolism and generates ROS while metabolizing xanthine into uric acid. Elevated ROS production due to increased uric acid levels can contribute to cardiovascular diseases and worsen HF [68,69,70]. High uric acid levels are associated with oxidative stress, inflammation and endothelial dysfunction, all of which are key factors in HF progression [69,71,72]. Elevated XO activity, resulting in increased uric acid production, is linked to HF development. XO-derived ROS contribute to endothelial dysfunction, inflammation and myocardial apoptosis, all of which are factors in HF pathophysiology [69]. Uric acid has emerged as a potential prognostic biomarker in HF. Elevated uric acid levels are associated with poorer outcomes in HF patients, including an increased risk of all-cause mortality, cardiovascular mortality and readmission [69]. Uric acid levels also correlate with symptom burden and can predict the severity of HF according to the New York Heart Association (NYHA) classification [71]. The Taiwan MJ cohort studies suggest that the impact of uric acid may differ between men and women with HF. In women, elevated uric acid levels are associated with a higher cardiovascular hazard ratio, which may be related to hormonal differences [69]. Uric acid levels are associated with central hemodynamic parameters in advanced HF patients, particularly the left-sided filling pressure (PCWP) and cardiac index (CI). This indicates that elevated uric acid levels are linked to altered hemodynamics in HF patients, potentially contributing to disease severity [70,71,72,73].

#### 6.2.4. Myeloperoxidase

Myeloperoxidase (MPO) is an enzyme produced during inflammation and oxidative stress by activated white blood cells. It significantly contributes to tissue damage in HF by inducing oxidative damage through the formation of reactive oxidants and radicals. These pathways significantly influence the development of coronary artery disease, vascular diseases and HF, contributing to atherogenesis, plaque vulnerability and ventricular remodeling [74,75]. In HF, increased MPO activity has been linked to various diseases, including atherosclerosis. Its association with vascular dysfunction, mediated by reduced NO availability and other mechanisms, contributes to impaired vasodilation, ATP depletion and increased purine metabolism [7]. Elevated MPO levels serve as a robust predictor of coronary artery disease risk. In chronic HF, these levels are strongly associated with HF prevalence, efficiently identifying HF patients. When combined with other markers such as C-reactive protein (CRP) and NT-proBNP, MPO levels significantly improve HF detection and correlate with severe HF and adverse clinical outcomes [7,74,76]. Chronic inflammation significantly impacts HF progression, particularly in HFpEF and HFrEF. HFpEF involves systemic and cardiac inflammation triggered by metabolic risk factors, activating endothelial cells and upregulating NOX2. Conversely, HFrEF may stem from inflammation initiated by myocardial infarction, sustained during tissue regeneration and remodeling, often involving immune cell infiltration [76,77]. Nitrosative stress, notably represented by the marker 3-nitrotyrosine (3-NT), is relevant in HFpEF’s pathogenesis. Elevated 3-NT levels, a result of reactive nitrogen species under oxidative stress, are observed in myocardial samples from HFpEF patients compared to HFrEF. MPO derived from leukocytes contributes to vascular dysfunction and cardiovascular diseases, including HFpEF [75,76]. Several studies exploring the role of MPO in chronic HF have shown that elevated MPO levels are correlated with uric acid levels in HFpEF patients. Additionally, MPO levels in HFrEF patients correlated with HF severity markers and were predictive of adverse clinical outcomes. While MPO shows promise as a diagnostic and prognostic marker in HF, its utility in the diagnosis of acute HF varies. Reichlin et al. suggest a potential role for MPO in risk management in acute HF, showing associations with 1-year mortality and long-term clinical outcomes, albeit with mixed diagnostic value compared to BNP [75,76,77]. Soluble NOX2, a significant contributor to oxidative stress, plays a crucial role in HF. It has been associated with heart conditions, particularly postoperative atrial fibrillation (POAF). Studies highlight the critical role of oxidative stress in POAF development, particularly the production of ROS such as superoxide (O^2−^), with NOX2’s inhibition showing potential in managingatrial fibrillation (AF) complications [7,78,79]. Moreover, increased levels of soluble NOX2-derived peptide were strongly associated with elevated serum isoprostane, a marker of oxidative stress, in persistent/PAF patients, suggesting a role for NOX2 in oxidative stress observed in POAF patients. In AF, NOX2-generated O^2−^ in the left atrial appendage (LAA) was associated with left atrial enlargement. Patients with POAF exhibited higher basal myocardial O^2−^ production, indicating increased oxidative stress, further supported by elevated urine F2-isoprostanes and isofurans [7,78,80]. Studies implicate NOX2 as a significant source of ROS in cardiovascular diseases, that is, ICM and AF, potentially leading to HF. Specifically, NOX2 in cardiomyocytes significantly contributes to ROS generation, especially in LVH, a crucial component in HF progression. Monitoring NOX2 activity, particularly through sNOX2-dp, shows promise in assessing cardiac remodeling post-myocardial infarction, contributing significantly to its pathogenesis [78,80,81]. Despite the potential of sNOX2-dp as a biomarker in HF and cardiovascular diseases, comprehensive research is essential to establish its diagnostic and prognostic value. Nonetheless, evidence consistently underscores the substantial contribution of NOX2 to oxidative stress in HF, suggesting the potential utility of sNOX2-dp as a valuable biomarker for monitoring oxidative stress levels in HF [79,80,81,82].

#### 6.2.5. 8-Hydroxyguanosine

8-Hydroxyguanosine (8-oxo(d)G) is an extensively studied product of nucleic acid oxidation formed by guanine oxidation, disrupting DNA pairing and causing gene mutations [79,83]. 8-oxoGsn, a metabolite reflecting oxidative damage to RNA, plays a significant role in heart disease through its association with oxidative stress [83,84,85,86]. Elevated levels of urinary 8-oxoGsn, indicative of RNA oxidative damage, have been linked to poor physical performance and cognitive impairment, suggesting a relationship between oxidative stress and a decline in these functions, which are also common in heart disease [83,84]. Repair mechanisms exist to mitigate the mutagenic effects of 8-oxoGsn, involving proteins, such as human polynucleotide phosphorylase (hPNPase), Y box binding protein 1 (Yb-1) and heterogeneous nuclear ribonucleoproteins (hnRNP), which silence and degrade damaged RNA [83,84]. Additionally, ATM serine/threonine kinase, a protein known for its role in the DNA damage response, may participate in the cellular response to RNA oxidative damage, although the precise mechanism has not been fully understood [83]. The metabolite of 8-oxo(d)G, known as 8-oxoGsn, is detected in urine and serves as a biomarker for oxidative damage. Si-Min Yao et al. demonstrated that elevated levels of urinary 8-oxoGsn were associated with aging-related diseases, suggesting a potential link with heart disease and poorer prognosis. Therefore, urinary 8-oxoGsn may serve as a valuable biomarker for assessing oxidative damage and evaluating the progression of heart disease. However, further research is needed to confirm these findings and establish causality, as well as to identify associated risk factors and determine the utility of urinary 8-oxoGsn in clinical practice for managing heart disease [85,87].

#### 6.2.6. F2-Isoprostanes

F2-Isoprostanes (IsoPs), generated from arachidonic acid breakdown, are detected in various cellular contexts, notably in monocytes during atherosclerosis driven by oxidative stress [88,89,90,91]. Elevated levels of IsoPs have been observed in the urine of individuals with unstable angina, indicating their potential as biomarkers for prognosticating complications in nonfatal myocardial infarction and HF progression [88]. IsoPs are primarily derived from tissue phospholipids and undergo hydrolysis to their free acid form, with subsequent metabolism, predominantly in the liver, and excretion in urine [90,91]. However, interpreting changes in unmetabolized IsoPs from urine requires caution due to potential disproportionate increases in renal disease. Reliable measurements of IsoPs involve stable isotope dilution mass spectrometry, which offers advantages over immunoassays by ensuring accuracy and minimizing variability [90]. IsoPs have been linked to coronary heart disease risk factors, disease extent and outcomes, supporting their role in atherogenesis. Elevated IsoP levels correlate with the severity of coronary heart disease and predict adverse cardiac events, suggesting their utility as prognostic markers. Heravi et al. analyzed associations between traditional cardiovascular disease risk factors and urinary and plasma isoprostane concentrations, which indicate systemic oxidative stress. The investigators found that different assays exhibited varying levels of effectiveness and sometimes contradictory correlations with specific cardiovascular disease risk factors. This implies that the type of body fluid analyzed and the isoprostane entity measured can influence the observed associations. Therefore, it is recommended to view isoprostanes as complementary rather than interchangeable in assessing oxidative stress. Additionally, while urinary isoprostanes are convenient, adjusting for urine creatinine levels could lead to significant confounding when comparing groups with different serum creatinine concentrations. The study also found consistently higher isoprostane concentrations in women, smokers, sedentary individuals and those with higher body mass index (BMI) and plasma triglycerides, suggesting greater oxidative stress and a potentially increased cardiovascular disease risk in these groups [89].

#### 6.2.7. Nitrotyrosine

Epidemiological studies have identified correlations between cardiovascular disease and the levels of 3-nitrotyrosine (NT), a marker of nitrosative stress, and asymmetric dimethylarginine (ADMA), an endogenous inhibitor of NO synthase [76,92,93]. NT is a marker of nitrosative stress resulting from the nitration process, affecting tyrosine residues in plasma proteins and atherosclerotic lesions. NT levels have been observed to rise in various inflammatory conditions, including peripheral thrombosis (PT) [92]. ADMA serves as an endogenous inhibitor of NO synthase, leading to enzyme uncoupling and subsequent peroxynitrite production, thereby inducing nitrosative stress in vascular endothelial and smooth muscle cells [76,92]. Oxidative and nitrosative stress can trigger immune responses and inflammasome activation, exacerbating inflammation and contributing to cardiovascular disease progression [92,93]. Ferlazzo et al. evaluated the inflammatory status by analyzing the transcription levels of inflammasome-associated genes in peripheral blood mononuclear cells (PBMCs) from recruited subjects. The results highlight significant alterations in the levels of ADMA, NT and Coenzyme Q10 (Co Q10) in individuals with PT, coronary artery disease (CHD) or both conditions, suggesting their potential as biomarkers for endothelial dysfunction and inflammation [92].

Nitrosative stress, evidenced by increased levels of 3-NT, plays a pivotal role in HFpEF pathogenesis, affecting the Inositol-Requiring Enzyme 1α-X-Box Binding Protein 1 (IRE1α-XBP1) pathway [76]. Elevated circulating 3-NT levels are associated with increased fibrosis and structural remodeling of the myocardium, highlighting their potential role as a diagnostic marker and therapeutic target [76,92,93].

Mommot et al. investigated circulating levels of MPO and 3-NT in HFpEF, HFrEF and healthy individuals. They found significantly elevated 3-NT levels in HFpEF compared to other groups, suggesting a distinct pathogenic mechanism involving oxidative/nitrosative stress in HFpEF. However, MPO levels did not significantly differ between groups, challenging previous findings associating MPO with HFpEF [76].

#### 6.2.8. NOX2-Derived Peptide

A novel approach to assessing NOX activity has emerged, involving the detection of a soluble NOX2-derived peptide (sNOX2-dp). This peptide, originating from the extramembrane portion of NOX2, is measurable in serum and plasma using the ELISA method, facilitating its clinical application. Initial investigations indicate that sNOX2-dp primarily originates from blood cells, with a minor contribution of endothelial cells [7,94,95]. However, comprehensive studies on its stability, metabolism and clearance are still lacking, limiting its widespread use. Despite this, there is growing interest in exploring sNOX2-dp as a potential biomarker, particularly in cardiovascular diseases such as HF [96,97].

In HF, NOX activity, especially in cardiomyocytes, is a significant source of ROS, contributing to pathological processes such as LVH and cardiac remodeling. Studies have demonstrated elevated NOX activity in failing myocardium, suggesting its potential as a therapeutic target and a prognostic indicator. While NOX2 appears to play a pivotal role in HF pathogenesis, the involvement of other NOX isoforms, i.e., NOX4, remains less understood and requires further investigation [94,95,96].

Furthermore, NOX2 has been implicated in AF, where increased NOX2 activity contributes to atrial remodeling and oxidative stress. Elevated levels of sNOX2-dp have been observed in AF patients, indicating its potential utility as a diagnostic marker in this context.

Overall, while the diagnostic and prognostic value of sNOX2-dp in HF and AF requires further validation through larger prospective studies, emerging evidence suggests its promising role as a biomarker for assessing oxidative stress and disease progression in cardiovascular conditions [7,97].

#### 6.2.9. Bilirubin

Liver dysfunction is prevalent in acute HF and chronic HF patients, contributing to adverse outcomes. The albumin–bilirubin (ALBI) score, initially developed for liver disease assessment, holds promise in predicting prognosis in HF patients [98,99,100]. The ALBI score integrates serum albumin and total bilirubin levels and has shown promise in predicting prognosis in patients with hepatocellular carcinoma. In HF, albumin and bilirubin have been identified as prognostic factors [98,101]. A study by Takayuki Kawata et al. suggested that the ALBI score at admission could predict in-hospital mortality in acute HF patients, with a high ALBI score correlating with increased mortality [98,99]. Hypoalbuminemia, common in HF, serves as a predictor for short-term prognosis, influenced by factors such as nutritional intake and systemic inflammation. Liver dysfunction associated with acute HF contributes to hypoalbuminemia and elevated serum bilirubin levels. The ALBI score, reflecting liver congestion and dysfunction, may provide insights into HF prognosis. Despite its origin in hepatocellular carcinoma, application of the ALBI score can be extended for assessment of liver dysfunction in various conditions, including HF. Similar to the best-established prognostic indicators, it can offer valuable information for patient management in acute HF admissions; however, large-scale studies are needed for validation of the relationships. Studies demonstrate associations between liver function markers and HF outcomes, with high bilirubin levels correlating with poorer prognosis [98,99,101].

Furthermore, the ALBI score remains independently associated with mortality in chronic HF patients, suggesting its utility as a risk assessment tool [101,102]. In ST-elevation MI (STEMI) patients undergoing primary percutaneous coronary intervention (PCI), elevated bilirubin levels are independently associated with an increased risk of in-hospital Major Adverse Cardiac Events (MACEs), indicating its potential as a predictive biomarker for adverse cardiac events [102]. However, further research is warranted to explore long-term outcomes and the impact of bilirubin levels post-discharge in STEMI patients. Overall, these findings emphasize the importance of liver function markers, particularly bilirubin, in assessing and managing HF and STEMI patients [101,102].

#### 6.2.10. Ceruloplasmin

Ceruloplasmin (Cp) is a liver-secreted protein crucial for maintaining copper balance in the body, binding to approximately 90% of plasma copper ions [103,104,105]. It exhibits antioxidant properties, acts as an oxidase and possesses ferroxidase activity, essential for iron metabolism. Clinical studies have linked elevated Cp levels with metabolic disorders, including metabolic syndrome, diabetes and cardiovascular diseases. Cp is an acute-phase reactant protein whose plasma concentration increases in response to inflammation, trauma or infection [106].

Smyła-Gruca et al. analyzed the association between oxidative/antioxidative balance markers and all-cause mortality in heart transplant candidates. The study included 154 patients with advanced HF who underwent qualification for heart transplantation. Markers such as Cp and catalase (CAT) showed excellent predictive power for identifying patients at high risk of death [103]. The study suggests that oxidative stress markers, particularly Cp and CAT, along with creatinine levels, could serve as valuable prognostic indicators in HF patients [103]. These findings highlight the importance of oxidative stress in HF progression and the potential utility of oxidative/antioxidative balance markers in the risk stratification and management of HF patients [103,104,105].

HFrEF involves complex physiological pathways, including oxidative stress, inflammation and iron deficiency [104]. Patients with HFrEF experience exercise intolerance due to reduced cardiac output, leading to symptoms such as dyspnea and fatigue [104]. Cp, an enzyme involved in iron metabolism and antioxidant activity, has emerged as a potential biomarker in HFrEF. Studies by Lazar-Poloczek et al. have shown associations between elevated Cp levels and an increased severity of HF, though its precise role in the disease process remains unclear [103,104,105]. The findings suggest that elevated Cp levels are associated with reduced exercise capacity and increased oxidative stress in HFrEF patients. Additionally, positive correlations were found between Cp levels and hepatic enzymes, suggesting a possible link between liver function and Cp metabolism. Overall, these results contribute to the understanding of the potential role of Cp as a biomarker in HFrEF and highlight its association with oxidative stress in the disease process. Further research is needed to elucidate the multifaceted role of Cp in heart failure pathology [104].

Romuk et al. investigated the prognostic value of Cp concentration, along with other common risk factors, in patients with HF [105]. The study demonstrated that patients with higher Cp levels, especially when combined with elevated NT-proBNP levels, faced an increased risk of all-cause mortality or heart transplantation within a year [105]. However, after adjusting for various factors, Cp alone did not remain a significant predictor. Notably, the combination of Cp and NT-proBNP concentrations showed better predictive accuracy than either biomarker alone. Other factors, such as left ventricular ejection fraction (LVEF), peak oxygen consumption (peak VO2) and certain treatments, were also identified as independent predictors of outcome. The study highlights the potential usefulness of Cp and NT-proBNP measurements in identifying high-risk HF patients [105].

## 7. Conclusions

Novel biomarkers in HF may improve diagnosis and prognosis. A multi-marker strategy is beneficial and can increase diagnostic accuracy and improve risk stratification in HF. The problem lies in choosing the most appropriate panel of markers for effective HF prevention and early diagnosis. It seems a difficult task to identify a panel of markers that characterized by sufficient sensitivity and specificity, which is the most important requirement for a new test to be clinically useful. Therefore, researchers should continue to look for the most effective markers. 

## Figures and Tables

**Table 1 medicina-60-00760-t001:** Oxidative stress and inflammatory parameters in heart failure.

Biomarker	Rutine Used in Laboratory Medicine	Availability of Ready Kits	Methods
Apelin	NO	YES	ELISA
FGF-23	NO	YES	ELISA/CLIA
Galectin 3	NO	YES	ELISA/CLIA
GDF 15	YES	YES	ELISA/CLIA
Klotho	NO	YES	ELISA
Collagen	YES	YES	ELISA/CLIA
MDA	YES	YES	COLORIMETRIC/FLUOROMETRIC/ELISA
sST2	NO	YES	ELISA
SH	NO	YES	COLORIMETRIC
Uric acid	YES	YES	COLORIMETRIC/FLUOROMETRIC
MPO	NO	YES	FLUOROMETRIC/SPECTROPHOTMETRIC/ELISA
Ceruloplasmin	YES	YES	ELISA
Bilirubin	YES	YES	COLORIMETRIC/ELISA
NOX2-derived peptide	NO	NO	WESTERN-BLOTT
Nitrotyrosine	NO	YES	ELISA
F2-Isoprostanes	NO	YES	ELISA
8-Hydroxyguanosine	NO	YES	ELISA

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
