# Peer review of "The Role of Oxidative Stress and Inflammatory Parameters in Heart Failure"

_medicina, 2024, doi:10.3390/medicina60050760_

Round 1

Reviewer 1 Report

Comments and Suggestions for Authors

Review of the Manuscript "Oxidative stress mechanism in heart failure: the role of biomarkers in HF prognosis"

The objective of the study was to describe oxidative stress mechanism in heart failure and the role of biomarkers in HF prognosis. The manuscript is well-written, with clear and concise language that enhances the accessibility of the content. The authors' ability to communicate complex ideas in a straightforward manner contributes significantly to the manuscript's readability and overall impact.

Having thoroughly examined the content, I would like to pose a few clarifying questions to better understand certain aspects of the study.

·         Abstract is too short and not informative.

·         Title cannot contain abbreviations.

·         What is missing here is how these biomarker can be determined in clinical labs?

·         Which of them could be determined?

·         Which are for clinical use and which of them are research parameters only?

·         What are the estimated costs of the biomarkers?

Kindly incorporate the responses within the manuscript to augment its overall quality.

Author Response

Dear reviewer,

Thank you for your valuable comments. Answers attached in the file

Reviewer 2 Report

Comments and Suggestions for Authors

The authors present a comprehensive analysis and examination of the involvement of oxidative stress in the initiation and advancement of HF. Furthermore, they emphasize specific oxidative stress and inflammatory biomarkers that hold significant clinical value in the context of HF. My comments are:

1. It is recommended that the title of the article be revised to enhance clarity and accuracy.

2. The abstract necessitates a more detailed revision as it currently lacks coherence.

3. Do you intend to discuss the utilization of oxidative stress biomarkers in predicting the prognosis of heart failure (HF) or HD biomarkers?

4. Are certain elements such as FGF23 and collagen considered oxidative markers?

5. Biomarkers can serve as indicators of various facets of HF pathophysiology, including oxidative stress, inflammation, cardiomyocyte apoptosis, matrix remodeling, and myocardial fibrosis. Hence, restructuring the manuscript based on these key pathophysiological events in HF is recommended.

6. It is preferable to elucidate the mechanism(s) through which oxidative stress impacts heart failure.

7. Acronyms should be provided for all terms upon their first usage.

8. It is suggested to include your search strategy in the manuscript and update your search not to miss any articles.

Comments on the Quality of English Language

The article would benefit from a thorough revision to enhance the quality of English, grammar, and verb tense consistency.

Author Response

Dear Reviewer,

Thank you for your valuable comments. Answers attached in the file

Reviewer 3 Report

Comments and Suggestions for Authors

The authors of this review extensively discussed common biomarkers utilized for heart failure detection in a narrative and highly informative manner. It would be helpful to highlight biomarkers that aid in the early detection of heart failure (HF). even before symptoms of heart failure become clinically apparent, enabling timely detection and intervention strategies. Could you please identify the biomarker that can be detected at an early stage to help detect heart failure?

Author Response

(The authors gave the same response as above.)
